# TEMPORAL PROMPTING MATTERS: RETHINKING REFERRING VIDEO OBJECT SEGMENTATION

## ABSTRACT

Referring Video Object Segmentation (RVOS) aims to segment the object referred to by the query sentence in the video. Most existing methods require end-to-end training with dense mask annotations, which could be computation-consuming and less scalable. In this work, we rethink the RVOS problem and aim to investigate the key to this task. Based on existing foundation segmentation models, we decompose the RVOS task into *referring*, *video*, and *segmentation* factors, and propose a ***Te**mporal Prompt Ge**ne**ration and Selec**t**ion (**Tenet**)* framework to address the referring and video factors while leaving the segmentation problem to foundation models. To efficiently adapt image-based foundation segmentation models to referring video object segmentation, we leverage off-the-shelf object detectors and trackers to produce temporal prompts associated with the referring sentence. While high-quality temporal prompts could be produced, they can not be easily identified from confidence scores. To tackle this issue, we propose *Prompt Preference Learning* to evaluate the quality of the produced temporal prompts. By taking such prompts to instruct image-based foundation segmentation models, we would be able to produce high-quality masks for the referred object, enabling efficient model adaptation to referring video object segmentation. Experiments on RVOS benchmarks demonstrate the effectiveness of the Tenet framework.

## 1 INTRODUCTION

Segmentation, as a fundamental task in computer vision, aims to partition images into distinct visual segments. With segmentation, individual image pixels would be categorized into specific regions or objects of interest, making it particularly essential for visual understanding in real-world applications, such as autonomous driving (Zendel et al., 2022), medical imaging (Shin et al., 2023), and robotics (Goyal et al., 2022). In traditional semantic segmentation (Long et al., 2015; Cheng et al., 2021), models are generally trained to classify objects within a limited set of pre-defined categories (*e.g.*, bear, gold fish, zebra, *etc.*). While promising results have been presented, these methods could not be easily applied in realistic scenarios, where the target objects are usually specified by free-form phrases or sentences (*e.g.*, "zebra eating grass with a goose in front of it") rather than the category names alone. To advance segmentation to handle such natural language descriptions, referring image segmentation (RIS) (Hu et al., 2016; Yang et al., 2022; Liu et al., 2023b;a) emerges to learn and understand complex text queries while associating them with the input images to produce segmentation masks for the target objects. Recent RIS works (Yang et al., 2022; Liu et al., 2023b;a) focus on designing Transformer-based or cross-attention models and employ vision-language learning to fuse the latent features from both modalities. Despite these advancements, when the visual input is a sequence of video frames rather than a single image, the time-varying object positions and appearance could result in inconsistent output masks from frame to frame, implying the inherent deficiency of image-based segmentation approaches.

Referring Video Object Segmentation (RVOS), in response to this, aims to segment the object referred to by a text query throughout the entire video. In contrast to RIS, RVOS is particularly faced with dynamic visual challenges, such as position and size variation, object occlusion or exit, pose deformation, and scene variation. Moreover, the referring sentences may contain long-term motions or actions (*e.g.*, "a gold fish on the left swimming towards the top right"), which could not be easily recognized from a single frame. To address such a challenging task, a number of works (Seo et al., 2020; Wu et al., 2022; 2023a; Miao et al., 2023; Tang et al., 2023; Luo et al., 2024; Wu et al.,

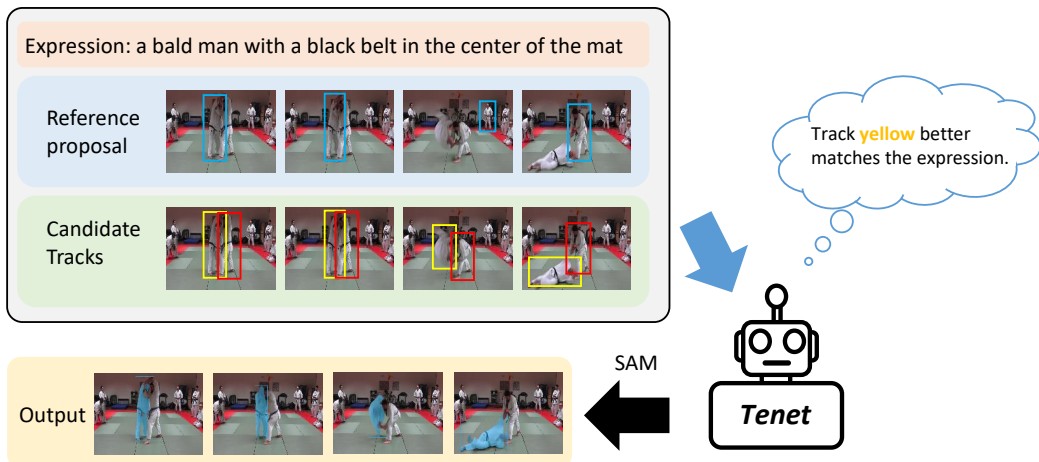

Figure 1: Given the expression, we first generate temporal prompts as the reference proposal and candidate tracks. Our proposed **Tenet** framework then selects the one that best aligns with the expression to prompt SAM, achieving referring video object segmentation.

2023b; Li et al., 2023a; Cheng et al., 2023a; Yan et al., 2024b; He & Ding, 2024; Yuan et al., 2024) have been presented. With the rapid development of Transformer (Vaswani et al., 2017), Refer-Former (Wu et al., 2022) takes the text inputs as queries to perform attention on the referred objects and links the corresponding queries across frames to achieve object tracking. Recent works like FS-RVOS (Li et al., 2023a) and OnlineRefer (Wu et al., 2023a) further extend RVOS to the few-shot setting and online pipeline to handle limited samples and ongoing videos in real-world scenarios, respectively. Nevertheless, most existing methods require end-to-end training for vision-language segmentation models, which could be computationally expensive and time-consuming. Moreover, the requirement of dense mask annotations for training impedes the scalability of these approaches.

Recently, several foundation segmentation models (Kirillov et al., 2023; Wang et al., 2023b; Zou et al., 2024) have been presented. Among them, SAM (Kirillov et al., 2023) is the most prominent one due to its overwhelming generalizability on various datasets. By employing large-scale model architectures and leveraging numerous image data for training, SAM can produce high-quality object masks according to visual prompts such as points or boxes, setting superior benchmarks for segmentation tasks. However, as SAM is trained solely with images and their associated masks, it could not properly handle natural language descriptions and video data in RVOS. Even though it is possible to incorporate additional grounding models (*e.g.*, Liu et al. (2024)) to generate text-associated box prompts and tracking models (*e.g.*, Cheng et al. (2023c)) to capture object movements across video frames, such naive combinations of off-the-shelf models has shown to be suboptimal (Li et al., 2023c), as they are individually trained for different tasks. This therefore raises a critical question: "*How to effectively and efficiently exploit foundation segmentation models for RVOS?*"

In this paper, we rethink the RVOS problem and aim to investigate the key to this challenging task. Based on the impressive results presented by foundation segmentation models, we decompose the RVOS task into the following three factors: *referring*, *video*, and *segmentation*, and focus on addressing the referring and video factors while leaving the segmentation problem to foundation models. To achieve this goal, we propose a **Temporal Prompt Generation and Selection (Tenet)** framework to efficiently adapt image-based foundation segmentation models to refer and segment video objects, as shown in Figure 1. Specifically, to generate visual prompts associated with the referring sentence, we leverage off-the-shelf object detectors and trackers to produce the **reference proposal** and **candidate tracks**. On the one hand, we perform object detection frame-by-frame to obtain box proposals indicating object positions, and the most confident (Top-1) proposals at each frame are together considered as reference proposals. On the other hand, to enhance the temporal consistency across frames, we also apply object tracking to Top-K proposals to form multiple candidate tracks. Based on our empirical analysis, a portion of these candidate tracks is shown to be superior to the reference proposal. However, such high-quality candidate tracks could *NOT* be

easily identified by the confidence scores of detected boxes. To tackle this problem, we propose *Prompt Preference Learning*, which employs a Transformer-based classifier to compare the quality of each candidate track with the reference proposal. By selecting the most preferred visual prompt to instruct image-based foundation segmentation models, high-quality masks for the referred object are produced, enabling efficient model adaptation to referring video object segmentation. Quantitative and qualitative experiments on standard RVOS benchmark datasets (Refer-Youtube-VOS and Refer-DAVIS$_{17}$) demonstrate the effectiveness of our proposed Tenet framework.

The contributions of this paper are summarized as follows:

- We rethink the RVOS problem and propose a *Temporal Prompt Generation and Selection (**Tenet**)* framework to efficiently adapt image-based foundation segmentation models to referring video object segmentation. Experiments on standard RVOS benchmarks demonstrate the effectiveness of the proposed Tenet framework.

- To generate visual prompts associated with the referring sentence, we leverage off-the-shelf object detectors and trackers to produce the ***reference proposal*** and ***candidate tracks***. Based on our empirical analysis, a portion of these candidate tracks is shown to be superior to the reference proposal.

- To identify high-quality candidate tracks, we propose ***Prompt Preference Learning*** to compare the quality of each candidate track with the reference proposal. By selecting the most preferred visual prompt to instruct image-based foundation segmentation models, we would be able to produce high-quality masks for the referred object.

## 2 RELATED WORKS

### 2.1 REFERRING IMAGE/VIDEO SEGMENTATION

Referring image segmentation (RIS) (Xu et al., 2023; Yang et al., 2022; Yu et al., 2023; Liu et al., 2023b) learns to segment the corresponding object in an image given a free-form text query. For example, PolyFormer (Liu et al., 2023b) re-formulates the RIS problem as a sequential polygon generation and then converts it to a segmentation mask. Also, PolyFormer further performs zero-shot transfer on the RVOS task to show its generalization ability for the video domain. However, the challenging issues for RVOS such as position and size variation, pose deformation, object occlusion, *etc*., may limit the performance of RIS methods.

Referring Video Object Segmentation (RVOS) (Wu et al., 2022; Han et al., 2023; Wu et al., 2023a; Miao et al., 2023; Tang et al., 2023; Luo et al., 2024; Wu et al., 2023b) strives to segment the object described by a free-form sentence query across the entire video duration. Recently, Refer-Former (Wu et al., 2022) views language as queries to pay attention to the referred object by adopting an encoder-decoder style in the transformer. However, this work only supports offline training and inference, limiting its usage in real-world scenarios. More recently, OnlineRefer (Wu et al., 2023a) further proposes an online RVOS setting to deal with the issues about offline limits, which makes it more possible to adapt to real-world scenarios. Nevertheless, most existing methods require end-to-end training for vision-language models, which could be computationally expensive and time-consuming. Moreover, the requirement of dense mask annotations for training impedes the scalability of those approaches. Instead, we propose to exploit foundation segmentation models without text- and temporal-aware prompting, which is trained without mask annotations and supports online settings. Very recently, several methods (Zhu et al., 2023; Lai et al., 2024; Yan et al., 2024a; Bai et al., 2024) are proposed to leverage the knowledge learned in large language models to address RVOS. Nevertheless, the results of these LLM-based methods are still inferior to traditional ones.

### 2.2 FOUNDATION SEGMENTATION MODELS

In recent years, foundation vision models have gained massive attention given their remarkable generalization capabilities on various downstream tasks. More recently, SAM (Kirillov et al., 2023) has introduced a foundation model specifically tailored for segmentation tasks. SAM allows specific position prompts (*e.g*., points, boxes, *etc*.) to demonstrate the zero-shot ability on the open vocabulary segmentation tasks with novel image distributions. Several works have studied the versatility of SAM, including remote sensing images (Chen et al., 2024; Wang et al., 2024), medical image

analysis (Ma et al., 2024; Chen et al., 2023; Wu et al., 2023c; Cheng et al., 2023b), and adaptation to video-based tracking task (Cheng et al., 2023c; Yang et al., 2023; Rajič et al., 2023), *etc*.

In addition to SAM, SegGPT (Wang et al., 2023b) and SEEM (Zou et al., 2024) have also emerged as generalized foundation segmentation models, showcasing comparable concepts. SegGPT exploits the concept of an in-context learning scheme to treat the classic segmentation problems as an in-context coloring problem. With this design, SegGPT is able to focus on more contextual information when training. On the other hand, SEEM extends the versatility of a single segmentation model by broadening the range of tasks. Similar to SAM, SEEM also supports various prompts including points, boxes, masks, *etc*. Specifically, SEEM proposes to align visual-semantic space to accommodate flexible multi-prompt input. However, both SegGPT and SEEM are not directly feasible for our RVOS task due to no specific adaptation to the video domain or enhancement of tracking ability.

For adaptation to tracking tasks with SAM, SAM-PT (Rajič et al., 2023) designs a point-based prompt enhancement for the original SAM point prompt to support classic video object segmentation tasks, while neglecting the importance of text prompt for advanced referring video object segmentation. Another example SAM-Track (Cheng et al., 2023c) attempts to utilize SAM for segmentation and detection of objects while the DeAOT (Yang & Yang, 2022) module captures the motion across frames for tracking the objects. On the other hand, SAM 2 (Ravi et al., 2024) introduces a memory attention mechanism on SAM to produce masklets for videos. Though it is possible to combine text-grounding detection models (*e.g.*, Grounding DINO (Liu et al., 2024)) with SAM-Track to tackle RVOS, RefSAM (Li et al., 2023c) has studied the possible concerns and indicates the unsatisfactory performance compared with current SOTAs in RVOS tasks. Different from the above, we propose temporal-aware prompting with foundation segmentation models (*e.g.*, SAM) to tackle RVOS problems.

## 3 PROPOSED FRAMEWORK: TENET

### 3.1 PROBLEM DEFINITION AND METHOD OVERVIEW

**Problem Definition.** For the sake of completeness, we first define the problem setting and notations used in this paper. In Referring Video Object Segmentation (RVOS), we assume that the training data contain a set of videos, where each video $V = \{I_t\}_{t=1}^{T}$ is a sequence of $T$ frames and the target object is associated with a referring sentence $S$. The goal of RVOS is to produce segmentation masks $M = \{M_t\}_{t=1}^{T}$ for the referred object from the video $V$ and referring sentence $S$. Since our goal is to prompt foundation segmentation models to achieve RVOS instead of training an end-to-end segmentation network as previous works (Wu et al., 2022; 2023a; Miao et al., 2023) did, we assume that we only have access to box-level annotations $GT = \{\hat{B}_t\}_{t=1}^{T}$ corresponding to the referred object as the ground truth, where each bounding box $\hat{B}_t = (\hat{x}_t, \hat{y}_t, \hat{h}_t, \hat{w}_t)$ is represented by the coordinate of the center point and the height and width. Such box-level annotations could be considered as a type of weak supervision for segmentation tasks (Khoreva et al., 2017).

**Method Overview.** With the impressive performance of foundation segmentation models such as SAM (Kirillov et al., 2023), we decompose the Referring Video Object Segmentation (RVOS) task into three core components: *referring*, *video*, and *segmentation*. We specifically address the challenges of the referring and video aspects, leveraging foundation models to handle segmentation. To this end, we propose a ***Temporal Prompt Generation and Selection (Tenet)*** framework to efficiently adapt image-based foundation segmentation models to referring video object segmentation. Specifically, our approach begins by utilizing off-the-shelf detectors and trackers to generate visual prompts corresponding to the referred object. We perform object detection frame-by-frame to obtain box proposals indicating object positions, and the most confident (Top-1) proposals at each frame are together considered as the ***reference proposal***. On the other hand, to enhance the temporal consistency across frames, we also form multiple ***candidate tracks*** by applying object tracking to Top-K proposals. To identify high-quality candidate tracks, we further propose ***Prompt Preference Learning***, which employs a Transformer-based classifier to compare the quality of each candidate track with the reference proposal. By selecting the most preferred visual prompt to instruct image-based foundation segmentation models, we would be able to produce high-quality masks for the referred object, enabling efficient model adaptation to referring video object segmentation. In the following subsections, we first introduce the generation process of temporal prompts and provide empirical

analysis in Section 3.2, and then detail the proposed Prompt Preference Learning to select temporal prompts in 3.3.

## 3.2 TEMPORAL PROMPT GENERATION AND ANALYSIS

Based on foundation segmentation models, we decompose the RVOS task into the following three factors: *referring*, *video*, and *segmentation*. To focus on the referring and video factors, we first investigate how to generate high-quality temporal prompts to benefit RVOS.

**Temporal Prompt Generation.** To obtain object positions, we consider Grounding DINO (Liu et al., 2024) as our detector to produce box proposals from the input video $V$ and the referring sentence $S$. Specifically, given the box annotations $GT = \{\hat{B}_t\}_{t=1}^T$ of the RVOS training data, we first finetune Grounding-DINO with the common regression loss and generalized IoU loss. Since there is typically only one target object in referring segmentation tasks, we simply select the output proposal with the highest confidence score at each frame to compute the loss instead of using the Hungarian algorithm (Carion et al., 2020) for box matching. We then use the pretrained and finetuned Grounding DINO for producing the reference proposal and candidate tracks, as described below:

- *Reference Proposal*: With the finetuned Grounding DINO, intuitively, we can simply take the most confident (Top-1) proposal at each frame $t$.
- *Candidate Tracks*: The above Top-1 proposals could be sensitive to prediction error and also inconsistent across frames. To achieve better temporal consistency, we consider the Top-K proposals and take an off-the-shelf motion-based tracker, OC-SORT (Cao et al., 2023), to perform object tracking. Here, the Top-K proposals are derived from the pretrained Grounding DINO since the finetuned one would lose generalizability and diversity and result in K repetitive boxes. As a result, we take the Top-K proposals from the pretrained Grounding DINO plus the Top-1 proposal from the finetuned one to produce candidate tracks from OC-SORT. Note that for the missed frames in each track, we use the Top-1 boxes to fill those untracked frames.

**Temporal Prompt Analysis.** For quantitative and quantitative analysis, we consider the aforementioned reference proposal, candidate tracks, and several of their variants and baselines. By taking them as visual prompts to SAM (Kirillov et al., 2023), we present the quantitative results in Table 1 and the associated visualization in Figure 2. Here, we use Refer-Youtube-VOS (Seo et al., 2020) as the training dataset, and since its validation set does not provide ground truth boxes, we use the Refer-DAVIS$_{17}$ (Khoreva et al., 2018) dataset for evaluation instead. For evaluation metrics, we consider mIoU for the box proposals, and we adopt the commonly used region similarity $\mathcal{J}$ (average IoU), contour accuracy $\mathcal{F}$ (average boundary similarity), and their mean $\mathcal{J}\&\mathcal{F}$ for the segmentation masks.

We detail each row in Table 1 as below: (a) Framewise Top-1 proposals from the pretrained Grounding DINO. (b) Framewise Top-1 proposals from the finetuned Grounding DINO. (c) The candidate track with the highest averaged confidence score produced by the Top-K proposals from the pretrained Grounding DINO plus the Top-1 proposal from the finetuned one. (d) The candidate track with the highest mIoU produced by the Top-K proposals from the pretrained Grounding DINO plus the Top-1 proposal from the finetuned one.

We list our important findings point-by-point as follows:

- *Prompting foundation segmentation models is a promising direction for RVOS.* By taking the ground-truth boxes to prompt SAM, we see that the resulting $\mathcal{J}\&\mathcal{F}$ is 83.6%, which is 15.6% higher than the state-of-the-art RVOS method, MUTR (Yan et al., 2024b).
- *The Top-1 proposals (reference proposal) from the finetuned detector outperforms the pretrained one.* Unsurprisingly, for the Top-1 proposals, the finetuned Grounding DINO (b) better fits the RVOS data and therefore surpasses the pretrained one (a) by 4.9% in $\mathcal{J}\&\mathcal{F}$.
- *The best candidate track produced from object tracking outperforms the reference proposal.* In (d), we present the result of the best-produced candidate track with the highest averaged box mIoU, and we see that the $\mathcal{J}\&\mathcal{F}$ is 5.6% higher than the reference proposal (b), showing that at least one of the candidate is of high quality.

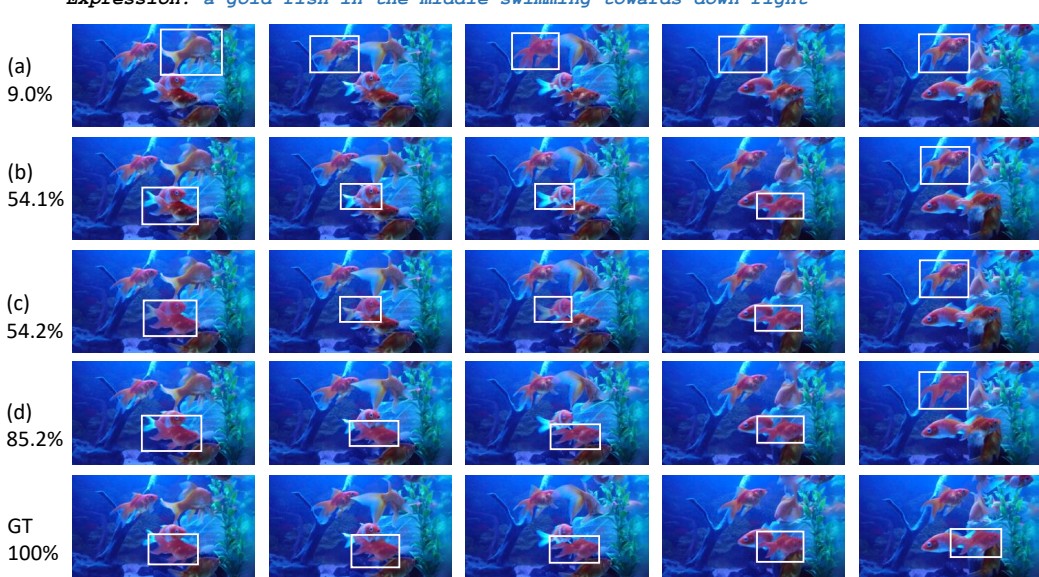

**Expression:** *a gold fish in the middle swimming towards down-right*

(a) 9.0%

(b) 54.1%

(c) 54.2%

(d) 85.2%

GT 100%

Figure 2: **Qualitative results when taking visual prompts derived from different methods to prompt SAM on the Refer-DAVIS$_{17}$ dataset.** Note that the score in the left is the box mIoU.

Table 1: **Quantitative results when taking visual prompts** derived from different methods to prompt SAM on the Refer-DAVIS$_{17}$ dataset.

| Method | Box mIoU | Ref-DAVIS17 $\mathcal{J}\&\mathcal{F}$ | $\mathcal{J}$ | $\mathcal{F}$ |
|---|---|---|---|---|
| (a) Reference Proposal (pretrained) | 71.2 | 65.2 | 62.3 | 68.0 |
| (b) Reference Proposal (finetuned) | 75.7 | 70.1 | 67.4 | 72.7 |
| (c) Candidate Track (highest conf.) | 72.7 | 68.9 | 66.0 | 71.7 |
| (d) Candidate Track (highest mIoU) | 81.8 | 76.2 | 73.0 | 79.5 |
| Ground-Truth Boxes | 100.0 | 83.6 | 80.1 | 87.2 |

- *High-quality candidate tracks could not be easily identified.* In (c), we present the result of the candidate track with the highest averaged confidence score, and we see that the $\mathcal{J}\&\mathcal{F}$ is significantly (7.3%) lower than the one with the highest box mIoU (d). This is because the frame-level confidence scores of detected boxes are not reliable for ranking video-level candidate tracks.

**Remarks.** In this section, we verify that prompting SAM is able to achieve superior performance and is a promising direction for referring video object segmentation. By performing object tracking to produce candidate tracks, we see that the best candidate would outperform the reference proposal from framewise detection. Nevertheless, high-quality candidate tracks could not be easily identified from confidence scores. As a result, we would like to learn a model that is able to properly evaluate the quality of candidate tracks.

### 3.3 PROMPT PREFERENCE LEARNING

As discussed in Section 3.2, high-quality candidate tracks could not be easily identified from confidence scores. To identify the visual prompts that best describe the referred object, we propose *Prompt Preference Learning* to compare the quality of each candidate track with the reference proposal, as shown in Figure 3. Specifically, to derive the visual representations for the objects indicated by the visual prompts, we adopt an image encoder to extract the latent features conditioned on the box proposals, resulting in $f^r$ and $f^c_i$ for the reference proposal and the $i$th candidate track, respectively. Along with the text feature $f^t$ derived by inputting the referring sentence into the text

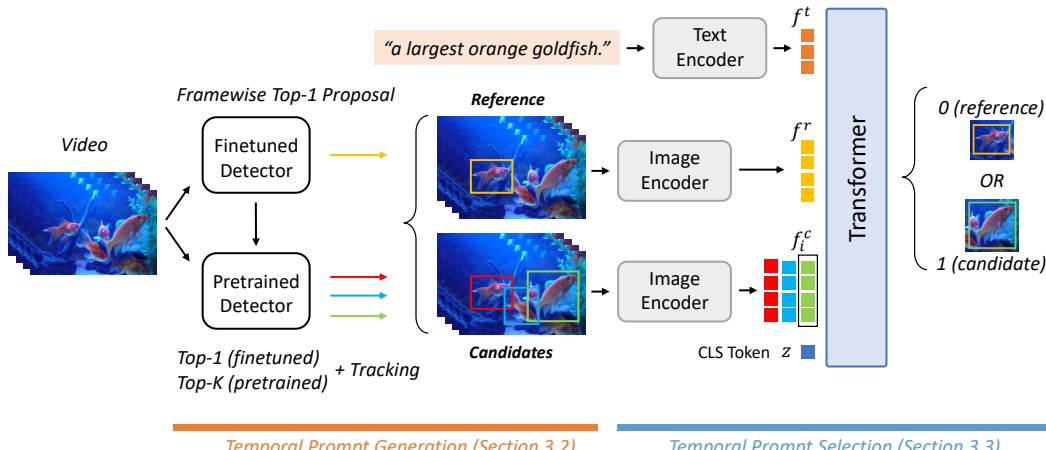

Figure 3: **Overview of the proposed Tenet framework.** We first produce the reference proposal and candidate tracks as described in Section 3.2, and then perform Prompt Preference Learning as detailed in Section 3.3.

encoder, we employ a Transformer-based binary classifier by taking visual features $f^r$ and $f_i^c$, the text features $f^t$, and an additional learnable classification token $z$ as input. Then, a standard binary cross entropy loss $L_{bce}$ is calculated as:

$$L_{bce} = -\sum_i \left[ y_i \log \sigma(s_i) + (1 - y_i) \log(1 - \sigma(s_i)) \right],$$

$$\text{where} \quad s_i = Transformer([z, f_i^c, f^r, f^t]).$$

(1)

Here, $\sigma(\cdot)$ denotes the sigmoid function and $y_i = 1$ if the $i$th candidate track is of higher mIoU than the reference proposal, otherwise 0. During inference, if there is at least one of the candidate track scores $\sigma(s_i)$ is over than 0.5, we select the candidate with the highest score. Otherwise, we select the reference proposal as the visual prompt.

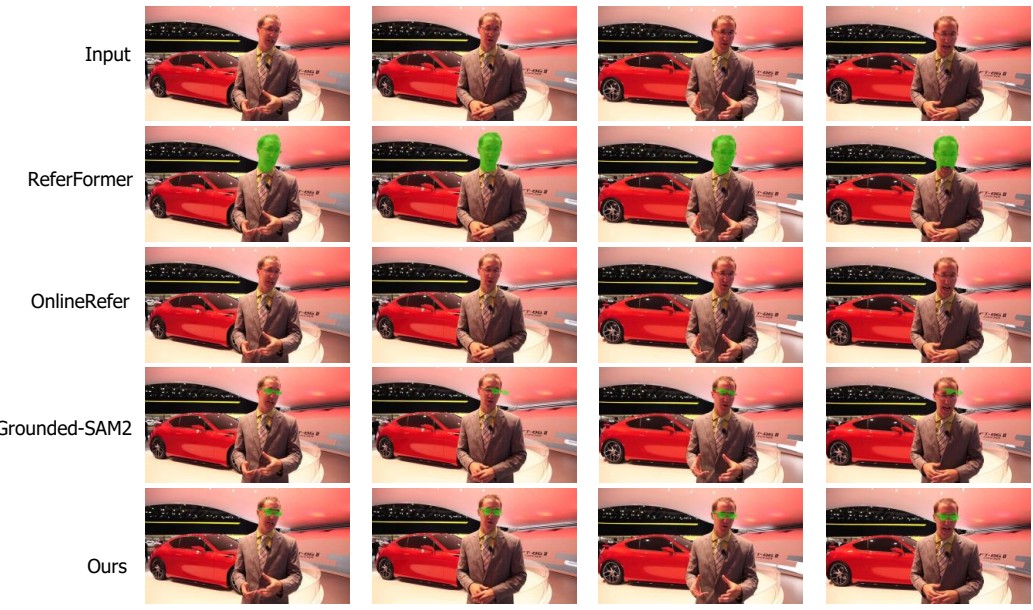

Figure 4: **Qualitative results on Ref-YouTube-VOS.**

Table 2: **Quantitative results on the validation split of Ref-YouTube-VOS and Ref-DAVIS17.** RefYT: Ref-YouTube-VOS, RefD: Ref-DAVIS, MeV: MeViS Ding et al. (2023), RefC: Ref-COCO (Mao et al., 2016; Yu et al., 2016), ReS: ReasonSeg (Lai et al., 2024), ReV: ReVOS (Yan et al., 2024a), YT: YouTube-VOS 2019 (Xu et al., 2018), D: DAVIS17 (Perazzi et al., 2016), O: Occluded VIS (Qi et al., 2022), LV: Long-term VOS (Hong et al., 2023), G: GOT-10K (Huang et al., 2019), La: LaSOT (Fan et al., 2019), T: TrackingNet (Muller et al., 2018), B: BDD100K (Yu et al., 2020), V: VIS19 (Yang et al., 2019), InT: InsTrack (Zhu et al., 2023), VC: Video-ChatGPT (Maaz et al., 2024), VD: large-scale video diffusion model (Wang et al., 2023a).

| Method | Publication | Referring & Video Training Data | Ref-YouTube-VOS | | | Ref-DAVIS17 | | |
|---|---|---|---|---|---|---|---|---|
| | | | $\mathcal{J}\&\mathcal{F}$ | $\mathcal{J}$ | $\mathcal{F}$ | $\mathcal{J}\&\mathcal{F}$ | $\mathcal{J}$ | $\mathcal{F}$ |
| *Standard RVOS approaches* | | | | | | | | |
| MTTR | CVPR'22 | RefYT | 55.3 | 54.0 | 56.6 | - | - | - |
| ReferFormer | CVPR'22 | RefC, RefYT | 62.9 | 61.3 | 64.6 | 61.1 | 58.1 | 64.1 |
| R$^2$-VOS | ICCV'23 | RefC, RefYT | 61.3 | 59.6 | 63.1 | - | - | - |
| HTML | ICCV'23 | RefC, RefYT | 63.4 | 61.5 | 65.2 | 62.1 | 59.2 | 65.1 |
| OnlineRefer | ICCV'23 | RefC, RefYT | 63.5 | 61.6 | 65.5 | 64.8 | 61.6 | 67.7 |
| SgMg | ICCV'23 | RefC, RefYT | 65.7 | 63.9 | 67.4 | 63.3 | 60.6 | 66.0 |
| TempCD | ICCV'23 | RefC, RefYT | 65.8 | 63.6 | 68.0 | 64.6 | 61.6 | 67.6 |
| RefSAM | arXiv'23 | RefC, RefYT | 62.1 | 60.9 | 63.3 | 69.5 | 65.9 | 73.2 |
| *Large-scale training approaches* | | | | | | | | |
| UniNEXT | CVPR'23 | RefC, RefYT, G, La, T, YT, B, V, O | 66.2 | 64.0 | 68.4 | 66.7 | 62.3 | 71.1 |
| DEVA | ICCV'23 | RefC, RefYT, YT, D, O | 66.0 | - | - | 66.3 | - | - |
| UniRef | ICCV'23 | RefC, RefYT, RefD, YT, O, LV | **67.4** | **65.5** | **69.2** | 66.3 | 62.9 | 69.7 |
| TrackGPT-7B | arXiv'23 | RefC, RefYT, ReS, InT | 56.4 | 55.3 | 57.4 | 63.2 | 59.4 | 67.0 |
| LISA-7B | CVPR'24 | RefC, ReS | 50.2 | 49.7 | 50.6 | 58.4 | 54.9 | 61.9 |
| VISA-7B | ECCV'24 | RefC, ReS, ReV, RefVY, RefD, MeV, VC | 61.5 | 59.8 | 63.2 | 69.4 | 66.3 | 72.5 |
| VD-IT-2B | ECCV'24 | RefC, RefYT, VD | 66.5 | 64.4 | 68.5 | 69.4 | 66.2 | 72.6 |
| VideoLISA-7B | NeurIPS'24 | RefC, ReS, RefYT, MeV, YT | 61.7 | 60.2 | 63.3 | 67.7 | 63.8 | 71.5 |
| *Efficient tuning approaches* | | | | | | | | |
| WRVOS | arXiv'23 | RefYT (box + 1st-frame mask) | 46.6 | 45.6 | 47.6 | 47.3 | 44.6 | 50.0 |
| Grounded-SAM | arXiv'24 | RefC (box) | 62.3 | 61.0 | 63.6 | 65.2 | 62.3 | 68.0 |
| Grounded-SAM2 | - | RefC (box) | 64.8 | 62.5 | 67.0 | 66.2 | 62.6 | 69.7 |
| **Tenet (Ours)** | - | RefC (box), **RefYT (box)** | 65.5 | 64.1 | 66.9 | **71.0** | **68.2** | **73.8** |

## 4 EXPERIMENTS

### 4.1 DATASETS AND IMPLEMENTATION DETAILS

**Datasets.** We conduct experiments on RVOS benchmark datasets: Refer-Youtube-VOS (Seo et al., 2020) and Refer-DAVIS$_{17}$ (Khoreva et al., 2018). Refer-Youtube-VOS is a large-scale dataset for RVOS, with $3,975$ videos, $7,451$ objects, and $27,899$ expressions. Refer-DAVIS$_{17}$ is augmented from the popular video object segmentation dataset, DAVIS$_{17}$ (Caelles et al., 2018). It contains 90 videos (60 for training and 30 for testing) with more than $1,500$ expressions. Since the annotations of the Ref-Youtube-VOS validation set are not publicly released, we evaluate the results on the official server. As for Ref-DAVIS$_{17}$, we use the official code for evaluation.

**Implementation Details.** For the image encoder, we use the CLIP image encoder pretrained from Guo et al. (2024) followed by a two-layer MLP. As for the text encoder, we use the pretrained CLIP text encoder with a two-layer MLP. As for the Transformer, we use a one-layer Transformer encoder layer. We only train the MLP's and Transformer's parameters. We use the learning rate of 0.0001 and train for 50 epochs with the Adam optimizer. All models are implemented in PyTorch and trained on NVIDIA H100 GPUs.

### 4.2 QUANTITATIVE AND QUALITATIVE RESULTS

We compare our Tenet framework with the following three types of methods for the RVOS task:

- Standard RVOS approaches: MTTR (Botach et al., 2022), ReferFormer (Wu et al., 2022), R$^2$-VOS (Li et al., 2023b), HTML (Han et al., 2023), OnlineRefer (Wu et al., 2023a), SgMg (Miao et al., 2023), TempCD (Tang et al., 2023), and RefSAM (Li et al., 2023c).

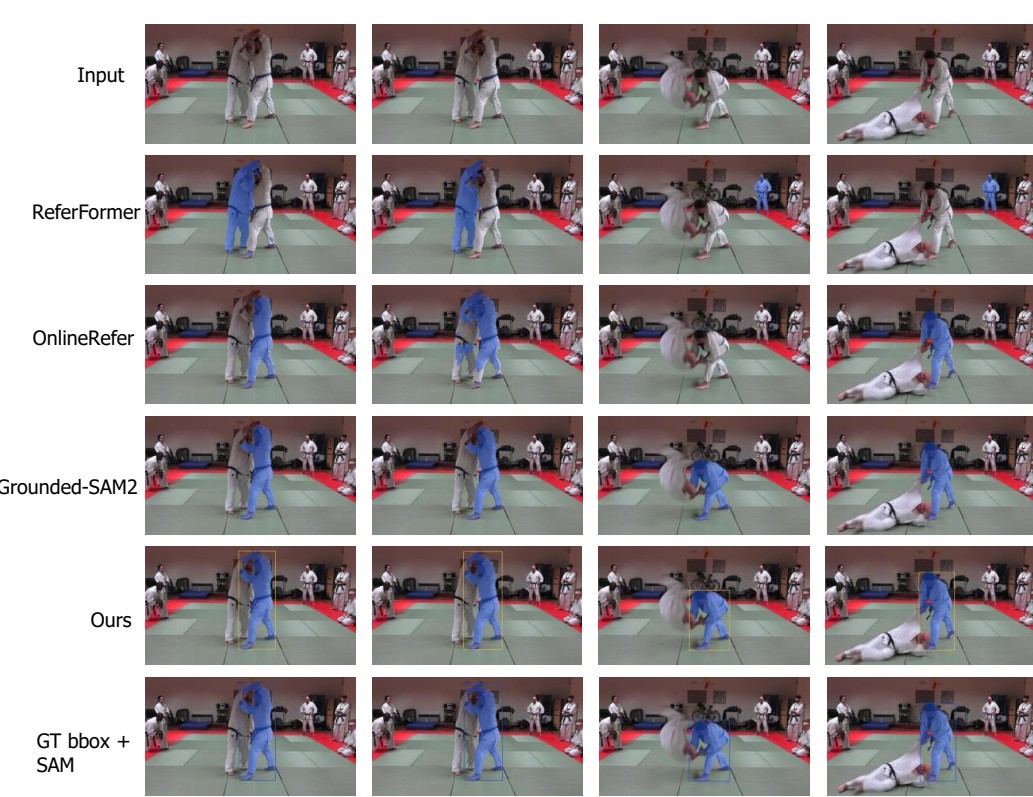

Figure 5: **Qualitative results on Ref-DAVIS17.**

- Large-scale training approaches: UniNEXT (Yan et al., 2023), DEVA (Cheng et al., 2023a), UniRef (Wu et al., 2023b), TrackGPT (Zhu et al., 2023), LISA (Lai et al., 2024), VISA (Yan et al., 2024a), VD-IT (Zhu et al., 2024), and VideoLISA (Bai et al., 2024).

- Efficient tuning approaches: WRVOS (Zhao et al., 2023), Grounded-SAM (Ren et al., 2024a), and Grounded-SAM2 (Ren et al., 2024b).

In Table 2, we first provide quantitative results on Ref-YouTube-VOS and Ref-DAVIS17. We see that our Tenet framework employs Prompt Preference Learning to select the track for prompting SAM, resulting in $65.5\%$ and $71.0\%$ in $\mathcal{J}\&\mathcal{F}$ on the two datasets, respectively. For the judo case in Figure 5, we see that Prompt Preference Learning is able to select the reference proposal for the referred object and produces similar results with the ground truth. As for the difficult eyeglasses case in Figure 4, our framework also performs better than Grounded-SAM2, demonstrating the effectiveness of our method.

Table 3: **Quantitative results of candidate track variants.** Note that $\star$ denotes that the candidate tracks are produced from top-5 proposals from the finetuned Grounding DINO.

| Method | Box mIoU | Ref-DAVIS17 | | |
|---|---|---|---|---|
| | | $\mathcal{J}\&\mathcal{F}$ | $\mathcal{J}$ | $\mathcal{F}$ |
| Candidate Track$^{\star}$ (highest mIoU) | 75.8 | 70.7 | 67.9 | 73.4 |
| Candidate Track (highest mIoU) | 81.8 | 76.2 | 73.0 | 79.5 |
| Candidate Track (merged) | 82.1 | 77.4 | 74.2 | 80.6 |

Table 4: **Hyper-parameter analysis when varying the number of proposals K.** Here, the candidate tracks are produced from the top-K proposals from the pretrained Grounding DINO plus the top-1 proposal from the finetuned one, and the result is the best candidate with the highest mIoU.

| Method | Box mIoU | Ref-DAVIS17 | | |
|---|---|---|---|---|
| | | $\mathcal{J}\&\mathcal{F}$ | $\mathcal{J}$ | $\mathcal{F}$ |
| Candidate Track (Top-2 (pretrained)) | 79.1 | 73.0 | 70.0 | 76.0 |
| Candidate Track (Top-3 (pretrained)) | 80.6 | 74.9 | 71.8 | 78.0 |
| Candidate Track (Top-5 (pretrained)) | 81.8 | 76.2 | 73.0 | 79.5 |
| Candidate Track (Top-10 (pretrained)) | 81.5 | 76.7 | 73.7 | 79.7 |

## 4.3 Ablation Studies

**Candidate Track Variants.** In Table 3, we additionally provide the results of candidate track variants. By comparing the first and second rows, we see that if we instead use Top-5 proposals from the finetuned detector to produce tracks, a $5.5\%$ performance drop in $\mathcal{J}\&\mathcal{F}$ would be observed. This is because the finetuned detector would lose diversity and produce repetitive box proposals, making the subsequent tracking meaningless. As a result, we take the Top-5 proposals from the pretrained Grounding DINO plus the Top-1 proposal from the finetuned one to produce candidate tracks from OC-SORT. Instead of simply selecting one best candidate track, another alternative is to greedily select and merge the tracks according to the mIoUs. Nevertheless, the resulting $\mathcal{J}\&\mathcal{F}$ is only $1.2\%$ higher in the third row. Hence, we simply consider the candidates individually.

**Number of Proposals.** In Table 4, we additionally provide the results when varying the number of proposals K from the pretrained Grounding DINO, and we see that the $\mathcal{J}\&\mathcal{F}$ saturates when K$\geq$5. Therefore, we simply set K to 5 for efficiency.

Table 5: Efficiency comparisons with recent RVOS methods, along with the $\mathcal{J}\&\mathcal{F}$ scores on Ref-YouTube-VOS and Ref-DAVIS17.

| Method | # of trainable parameters | Ref-YTVOS | Ref-DAVIS |
|---|---|---|---|
| ReferFormer (Wu et al., 2022) | $\sim$112M | 62.9 | 61.1 |
| OnlineRefer (Wu et al., 2023a) | $\sim$221M | 63.5 | 64.8 |
| DEVA (Cheng et al., 2023a) | $\sim$112M | 66.0 | 66.3 |
| Tenet (**Ours**) | $\sim$45M | 65.5 | 71.0 |

**Efficiency Comparisons.** In Table 5, we also provide efficiency comparisons with recent works. We see that the number of trainable parameters of our method is over 2 times fewer than DEVA. This is because our proposed Tenet framework learns to prompt foundation models for efficient adaptation instead of training a vision-language model end-to-end. Together with the quantitative comparisons in Table 2, we validate that our proposed Tenet framework is preferable in terms of performance, setting, and efficiency.

## 5 Conclusion

In this paper, we rethink the RVOS problem and decompose the RVOS task into the following three factors: *referring*, *video*, and *segmentation*, and propose the Tenet framework to address the referring and video factors while leave the segmentation problem to foundation models. To efficiently adapt image-based foundation segmentation models to referring video object segmentation, we leverage off-the-shelf object detectors and trackers to produce the reference proposal and candidate tracks, which would serve as temporal prompts to instruct foundation segmentation models to produce object masks. With the proposed Prompt Preference Learning, we would be able to identify temporal prompts sutiable for prompting SAM, enabling efficient model adaptation to referring video object segmentation. Experiments on standard RVOS benchmarks demonstrate the effectiveness of the proposed Tenet framework.

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
