# OpenReview forum: "Temporal Prompting Matters: Rethinking Referring Video Object Segmentation"
_ICLR.cc/2025/Conference — ICLR 2025 Conference Withdrawn Submission_

### Official Review · Reviewer_5WbH · 2024-10-28

**Soundness:** 2
**Presentation:** 2
**Contribution:** 2
**Rating:** 5
**Confidence:** 5

**Summary:**

This paper proposes to decouple the task of referring video object segmentation (RVOS) into three sub-tasks referring, video, and segmentation. Several foundation models like Grounding DINO and SAM are used here. In specific, the proposed solution is a three-stage solution. Firstly, it employs Grounding DINO to generate frame-wise proposal boxes or candidate tracks. Secondly, it introduces “prompt preference learning” to select the high-quality candidate. Thirdly, SAM bases the candidate to produce a pixel-wise segmentation mask. The experiments are evaluated on two benchmarks, Refer-DAVIS17 and Refer-Youtube-VOS.

**Strengths:**

- The proposed method employs current foundation models to address the task of RVOS, whose idea is straightforward.

- This paper is well-written and easy to follow.

**Weaknesses:**

The paper has some obvious shortcomings as follows.

---
- **The multi-stage inference solution is not novel, because it was used in one early study [1].** This work [1] produces multiple candidate object tracks and then selects the most matched one for pixel-wise segmentation. However, this kind of solution with multi-stage inference is hard to deploy and replaced by end-to-end solutions in recent years. Despite employing sophisticated foundational models to generate candidate tracks in this paper, the core concept of their proposal-selection process remains unchanged. As a result, the in-depth exploration of the paper "Rethinking Referring Video Object Segmentation" is not enough.

      [1] Video Object Segmentation with Referring Expressions, ECCV2018.

---

- **The design motivation of the two proposal generation methods (reference proposal and candidate tracks) are somewhat unclear.** Logically, the candidate tracks would exhibit greater temporal consistency, and, consequently, better segmentation performance compared to frame-wise reference proposals. The data in Table 1 also support this.  What is the primary reason or necessity for introducing frame-wise reference proposals?

---

- **The experiment results are incomplete.**
     1. The large-scale motion-based RVOS benchmark, MeViS[2], has not been evaluated. This benchmark is more focused on leveraging motion cues in language instructions, which is more challenging yet effective for evaluating RVOS models.
     2. Why are powerful MLLMs (like GPT-4o) not included in comparing the best candidate track identification?
     3. In addition to Grounding DINO + SAM2, another powerful model, Florence-2 + SAM2 [3], is not included.


      [2] MeViS: A Large-scale Benchmark for Video Segmentation with Motion Expressions, ICCV2023.
      [3] https://github.com/IDEA-Research/Grounded-SAM-2

---

In summary, this paper on revisiting the RVOS task lacks enough in-depth insight, clear motivations for its proposal generation methods, and detailed experiments. These problems lead to concerns about the claimed effectiveness of the proposed method.

**Questions:**

- Why does the proposed method, as shown in Table 5, have a small number difference in training parameters despite employing only one Transformer decoder layer, while the compared methods use a total of backbone models, six encoders and six decoders? If the parameter count is not six times larger in the compared methods, what factors contribute to this discrepancy?

---

### Official Review · Reviewer_gkZA · 2024-11-02

**Soundness:** 3
**Presentation:** 3
**Contribution:** 2
**Rating:** 3
**Confidence:** 4

**Summary:**

The paper proposes Tenet framework for Referring Video Object Segmentation. This framework decomposes the RVOS task into referring, video, and segmentation factors. By using object detectors and trackers to generate temporal prompts associated with query sentences, Tenet adapts image-based foundation segmentation models to RVOS. One weighting module based on Transformer is also proposed to better select candidate tracks.Experimental results demonstrate the effectiveness of Tenet.

**Strengths:**

* Figures 1 effectively elucidate the core concepts of this paper.
* The paper is articulate and easy to follow.

**Weaknesses:**

1. A notable concern is that this paper reads more like an engineering report than a research contribution. The primary question posed, as stated in the introduction, revolves around how to effectively and efficiently leverage foundational segmentation models for RVOS. This question might be perceived as straightforward by the vision-language and video-language modeling communities, potentially categorizing existing challenges as engineering rather than research problems.
2. The approach of initially detecting objects at the image level, propagating object masks or tracking object boxes to the video level, and subsequently weighting candidate tracks with a Transformer-based module has been extensively explored and proven effective in existing literature [ref-1]. Consequently, the technical novelty of this paper appears limited.
3. Incorporating a video-level foundational model like SAM-Track [ref-2], which not only segments but also tracks objects at the video level, might simplify the problem. In such a scenario, the only remaining challenge would be to weigh the candidate tracks, presenting an alternative avenue for the authors to explore.

[ref-1] Rethinking cross-modal interaction from a top-down perspective for referring video object segmentation. In CVPRW, 2021. Champion solution in YouTube-VOS 2021 RVOS track.

[ref-2] Segment and Track Anything.

**Questions:**

It is recommended to include an additional figure to visually depict the pipeline of the proposed approach, offering readers a clearer understanding of the methodology.

---

### Official Review · Reviewer_Webj · 2024-11-02

**Soundness:** 2
**Presentation:** 3
**Contribution:** 2
**Rating:** 3
**Confidence:** 4

**Summary:**

This paper aims to solve the referring video object segmentation task, which decompose the task into referring, video, and segmentation factors. Specifically, it focuses on addressing the referring and video factors while leaving the segmentation problem to foundation models. It uses off-the-shelf object detectors and trackers to produce the reference proposal and candidate tracks. A Transformer-based classifier is used to select the most preferred visual prompt (bounding box) to instruct image-based foundation segmentation model (i.e., SAM) to generate masks of referred object.

**Strengths:**

The proposed prompt preference learning method (a Transformer-based classifer) is effective to select visual prompt to guide the image-based foundation segmentation model to generate high-quality referred object. It produces the SOTA results on the Ref-DAVIS17 dataset, i.e., 71.0 T&F score.

**Weaknesses:**

1. It seems to combine multiple individual modules, such as Grounding DINO, the OC-SORT tracking algorithm to generate more accurate prediction of referred object to prompt the image-based foundation segmentation model SAM. The method is very straightforward and the contribution of this paper is limited.

2. Could you provide more thorough analysis of the reasons that the proposed method performs worse than UniNEXT [CVPR23], DEVA[ICCV23], UniRef [ICCV23], VD-IT-2B[ECCV24] on the Ref-YouTube-VOS dataset?

3. At L504, why the finetuned detector would lose diversity and produce repetitive box proposals? Could you provide more details of the finetuning process, e.g., how to collect the training data and how to determine the detailed parameters of the model training?

4. At L244, two duplicate "quantitative", i.e., "quantitative" -> "qualitative".

**Questions:**

Please see the weakness section.

---

### Official Review · Reviewer_cmdL · 2024-11-04

**Soundness:** 2
**Presentation:** 2
**Contribution:** 2
**Rating:** 3
**Confidence:** 4

**Summary:**

1.	In this paper, the authors decompose the RVOS task into referring, video, and segmentation factors, and propose a Temporal Prompt Generation and Selection (Tenet) framework to  address the referring and video factors while leaving the segmentation problem to foundation models
2.	Experiments on RVOS benchmarks demonstrate the effectiveness of the Tenet framework.

**Strengths:**

1.The decomposition of RVOS into referring, video, and segmentation factors is a unique approach that allows for targeted improvements in each area.

2.The paper provides extensive quantitative and qualitative results, demonstrating the effectiveness of the proposed framework over existing methods.

**Weaknesses:**

1.	Complexity: The Tenet framework involves multiple components, which might increase the complexity and is not scalable.
2.	The performance of the Tenet framework is heavily dependent on the quality of the pre-trained object detectors and trackers used. The paper could explore the impact of different foundation models on the performance of the Tenet framework.
3.	Ablation studies: More ablation studies could be conducted to understand the contribution of each component of the Tenet framework to the overall performance.
4.	Cost: The computational cost of training and inference the Prompt Preference Learning model and running detectors and trackers is not discussed.
5.	Lack of sufficient experimental results across various benchmarks, including MeVIS and ReVOS.
6.	Most of the methods compared are in the first half of 2023 and before, and some of the latest work is not compared [1,2].
[1]. SOC: Semantic-Assisted Object Cluster for Referring Video Object Segmentation
[2]. Referred by Multi-Modality: A Unified Temporal Transformer for Video Object Segmentation

**Questions:**

See weakness

---

### Official Review · Reviewer_Cafu · 2024-11-04

**Soundness:** 2
**Presentation:** 3
**Contribution:** 2
**Rating:** 5
**Confidence:** 4

**Summary:**

Different from previous models which directly output referent mask and is thereby trained using mask annotations, the authors propose to decompose RVOS into the referent box prediction part and leave the segmentation to pretrained box-promptable 2D SAM.

Specifically, Tenet first finetunes GroundingDINO using RVOS data in frame-wise manner. For each frame, Tenet generates multiple
 referent bounding box using the finetuned and pretrained GroundingDINO model. An off-the-shelf non-referring tracker is used to concate the box to temporal tube. To get the most probable tube candidate, the authors proposesPrompt Reference Learning to train a referring classifier. The chosen tube is taken as input to the 2D SAM to get the temporal referent segmentation. Tenet archives SOTA performance on Ref-DAVIS17.

**Strengths:**

1. The formulation of decomposing RVOS to referent box prediction and SAM segmentation is novel to RVOS. This view decomposes RVOS into referent filtering (vision-text information fusion) and vision-specific segmentation.

2. The authors demonstrate that using weak-annotation like boxes can achieve comparable and better performance for RVOS.

3. The paper is well written and demonstrate both qualitative and quantitative comparison.

**Weaknesses:**

1. The experiment lacks comparison on A2DS [2], JHDBS [2], and MeViS [1], which are common RVOS benchmarks.

2. It seems not general enough to decompose RVOS into box detection and promptable SAM segmentation. What if the sentence refers to multiple non-overlapping instances (boy and girl) or a semantic scattered region(sky, grass)? The authors should show the results on MeViS [1] (which includes samples for multiple instances referring) to validate the feasibility of Tenet on these circumstances.

3. As a R(Video)OS method, the modules related to video processing are the off-the-shelf Tracker and the Transformer classifier. Moreover, the Tracker does not take language as input. The new problem formulation seems also applicable for Referring Image Segmentation, what designs are novel contribution to RVOS?

[1] MeViS: A Large-scale Benchmark for Video Segmentation with Motion Expressions.
[2] Language as Queries for Referring Video Object Segmentation

**Questions:**

I can certainly improve my score if the author can provide more comprehensive experiments on A2DS, JHDBS, and MeViS. Moreover, the author should provide explanations on Weakness 2 to show Tenet's generalization ability.

---

### Note · Authors · 2024-11-27

I have read and agree with the venue's withdrawal policy on behalf of myself and my co-authors.